https://doi.org/10.1038/s42003-022-03505-7　　**OPEN**
# The INSIDEOUT framework provides precise signatures of the balance of intrinsic and extrinsic dynamics in brain states

Gustavo Deco [1,2,3,4✉], Yonatan Sanz Perl [1,5], Hernan Bocaccio[5], Enzo Tagliazucchi[5,6] & Morten L. Kringelbach [7,8,9✉]

Finding precise signatures of different brain states is a central, unsolved question in neuroscience. We reformulated the problem to quantify the 'inside out' balance of intrinsic and extrinsic brain dynamics in brain states. The difference in brain state can be described as differences in the detailed causal interactions found in the underlying intrinsic brain dynamics. We used a thermodynamics framework to quantify the breaking of the detailed balance captured by the level of asymmetry in temporal processing, i.e. the arrow of time. Specifically, the temporal asymmetry was computed by the time-shifted correlation matrices for the forward and reversed time series, reflecting the level of non-reversibility/non-equilibrium. We found precise, distinguishing signatures in terms of the reversibility and hierarchy of large-scale dynamics in three radically different brain states (awake, deep sleep and anaesthesia) in electrocorticography data from non-human primates. Significantly lower levels of reversibility were found in deep sleep and anaesthesia compared to wakefulness. Non-wakeful states also showed a flatter hierarchy, reflecting the diversity of the reversibility across the brain. Overall, this provides signatures of the breaking of detailed balance in different brain states, perhaps reflecting levels of conscious awareness.

[1] Center for Brain and Cognition, Computational Neuroscience Group, Department of Information and Communication Technologies, Universitat Pompeu Fabra, Roc Boronat 138, Barcelona 08018, Spain. [2] Institució Catalana de la Recerca i Estudis Avançats (ICREA), Passeig Lluís Companys 23, Barcelona 08010, Spain. [3] Department of Neuropsychology, Max Planck Institute for Human Cognitive and Brain Sciences, 04103 Leipzig, Germany. [4] School of Psychological Sciences, Monash University, Melbourne, Clayton, VIC 3800, Australia. [5] Department of Physics, University of Buenos Aires, Buenos Aires, Argentina. [6] Latin American Brain Health Institute (BrainLat), Universidad Adolfo Ibanez, Santiago, Chile. [7] Centre for Eudaimonia and Human Flourishing, Linacre College, University of Oxford, Oxford, UK. [8] Department of Psychiatry, University of Oxford, Oxford, UK. [9] Center for Music in the Brain, Department of Clinical Medicine, Aarhus University, Aarhus, Denmark. ✉email: gustavo.deco@upf.edu; morten.kringelbach@psych.ox.ac.uk

A fundamental, unsolved problem in neuroscience is how to answer the deceptively simple question of how to define and quantify a brain state, since such a signature could provide a Rosetta stone for decoding the fundamental laws of brain function[1–3]. To paraphrase William James' famous quote about attention, although 'everybody knows what a brain state is', we need a better theoretical framework to provide a distinguishing signature able to differentiate different brain states, which is currently the focus of intensive investigation in the field[1,4–10].

This previous research has made it clear that the most essential aspect of such a definition should aid in creating a framework for describing brain states in terms of underlying causal mechanisms and dynamical complexity. One of the most important steps in this research was undertaken by Massimini and colleagues who offered an elegant method for quantifying the dynamical complexity of brain states using perturbation-elicited variations in intrinsic global brain activity during very different brain states, including wakefulness, sleep, anaesthesia, and post-coma states[11–13]. They developed the perturbational complexity index (PCI), which distinguishes between distinct brain states by capturing significant changes in brain-wide spatiotemporal propagation of external stimulus[13]. In turn, this inspired the development of the Perturbative Integration Latency Index (PILI), a neuroimaging modelling-based measure characterising the recovery of perturbed brain dynamics to regain equilibrium after suppression of the perturbation[14,15].

Complementary to these dynamical complexity approaches that measure the intrinsic responses, here we hypothesise that a better signature could arise from measuring the differential effects of the extrinsic environment on the intrinsic brain dynamics in different brain states. This hypothesis stems from the well-established observation that the environment is driving sensory regions (lower in the brain hierarchy) much stronger than regions in the top of hierarchy, which are, in contrast, much more intrinsically driven. Indeed, the orchestration of a brain state must depend on the 'inside out' balance of intrinsic and extrinsic brain dynamics, which could therefore serve as a distinguishing signature of a brain state. This 'inside-out' perspective is inspired by the ideas put forward by Buzsaki, where the self-organized dynamics of the brain constrains how it acts on the world rather than being driven by sensations[16]. Here, however, we were only specifically interested in capturing the change in balance of intrinsic and extrinsic brain dynamics as measured in change in the hierarchy of the causal interactions in the intrinsic brain dynamics of different brain states. Capturing this 'inside out' balance has proven challenging, although recent results in thermodynamics suggest the temporal asymmetry of events, i.e. the arrow of time, could provide exactly the right tools for capturing the driving of the environment on a physical system like the brain[17,18]. It is well-known that survival in any living system is predicated on breaking the detailed balance of the transitions between the underlying microscopic states[19,20]. The fluxes of transitions between different states disappear in an equilibrium system with detailed balance[21–23]. Thermodynamics provides a convenient way of describing what happens when the fluxes of transitions vanish and become reversible in time in an equilibrium system[17]. In contrast, in a non-equilibrium system, where the balance is broken, the net fluxes between the underlying states become irreversible, establishing an arrow of time[18,24–27]. This idea of how non-equilibrium is intrinsically linked to non-reversibility[18] and the production of entropy, leading to the arrow of time, was originally proposed by Arthur Eddington[28] and has been widely studied in physics and biology[17,18,24–27].

We used these fundamental, theoretical insights to create the INSIDEOUT framework capable of capturing the 'inside out' balance of intrinsic (inside) and extrinsic (out) brain dynamics by directly estimating the arrow of time in brain signals. The main idea is to capture the asymmetry in temporal process by comparing time-shifted correlation matrices for the forward and reversed time series. This provides a quantification level of non-reversibility and consequently the degree of non-equilibrium in the brain dynamics of different brain states. This is a simpler alternative to estimating the arrow of time through computing the entropy production rate, which can be difficult to directly compute from brain signals, requiring several assumptions, although see recent research[21,22].

The INSIDEOUT framework allowed us to estimate the precise signatures of three radically different brain states (awake, deep sleep and anaesthesia) in ECOG brain data from non-human primates[29,30]. Crucially, we found significant different signatures for each brain state both in terms of the two measures of non-reversibility and hierarchy computed from the large-scale dynamics. Overall, this measure of the arrow of time in brain signals provides a direct quantification of the 'inside out' balance of intrinsic and extrinsic brain dynamics. As such this is a signature of conscious awareness, of how the environment is differentially driving the brain dynamics out of equilibrium depending on the underlying brain state.

## Results

The INSIDEOUT framework was inspired by a key idea, introduced in[18], namely capturing the temporal asymmetry of a dynamical system by extracting both the forward timeseries and its time reversed version (Fig. 1). Comparing the distinguishability between these two timeseries, we can extract the level of non-reversibility and consequently the level of non-equilibrium, i.e. how the external, extrinsic environment drives the internal, intrinsic dynamics. In other words, when the forward and reversed timeseries are not distinguishable, the system is reversible and in equilibrium - whereas when the level of distinguishability increases, the system becomes more and more non-reversible and closer to non-equilibrium. In this way, the change in causal interactions in brain dynamics can be quantified for different brain states.

Specifically, the important components of the INSIDEOUT framework are shown in Fig. 1, from the fundamental physics to the extraction of forward timeseries and construction of the time-reversed timeseries from the brain data. The principle of the arrow of time is then estimated from pairwise comparison of two timeseries. This relies on constructing their time-reversed versions and characterising the asymmetry in time through the shifted correlation. As can be seen, the shifted correlation decays more rapidly for signals with weak compared to strong time dependency (see details below). Finally, this fundamental principle is generalised to multivariate large-scale brain signals.

More specifically, Fig. 1a shows an example from thermodynamics of a physical system driven to non-equilibrium by external forces[18]. On the left is shown a Brownian particle in a moving potential, whose position is controlled by the extrinsic environment, driving the potential from one position to another in the forward process. The graph (on the top right) plots examples of forward evolving trajectories (light grey lines) with their average (solid grey line). Note how the average trajectory lags behind the centre of the potential (grey dashed line), i.e., the time evolution of the vertex of the parabola. In contrast, the graph at the bottom right plots the same but now for the sample reversed trajectories (light red lines) and their average (solid red line). Here, the average reversed trajectory leads the potential's centre (red dashed line). Computing the differences between the lines for average and the potential's centre for both forward and

**Time's arrow in physical system and in the brain**

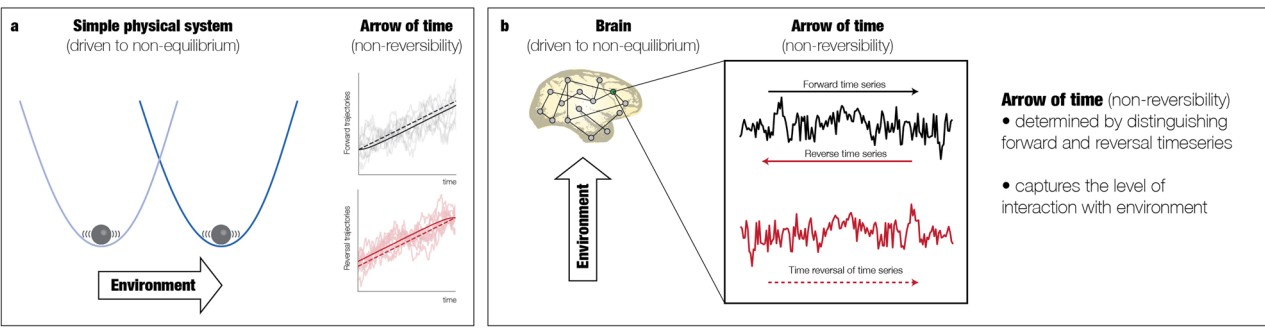

**Pairwise irreversibility of forward and reversal timeseries**

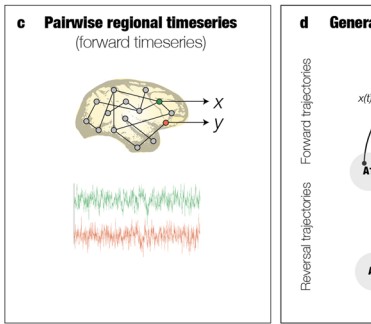
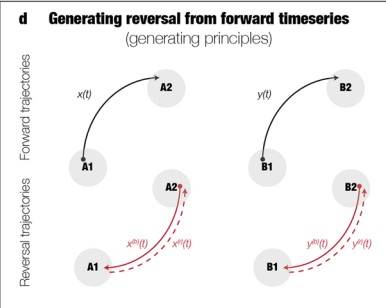
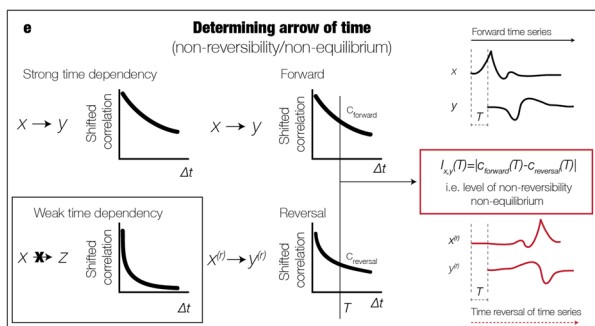

**Whole-brain irreversibility of forward and reversal timeseries**

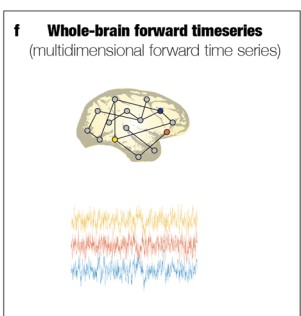
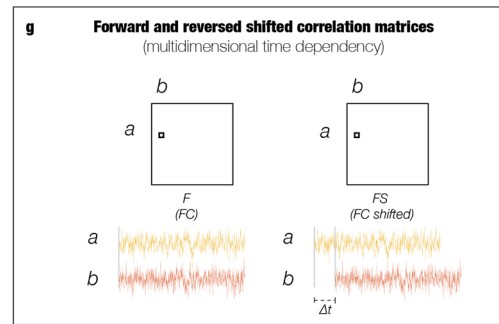
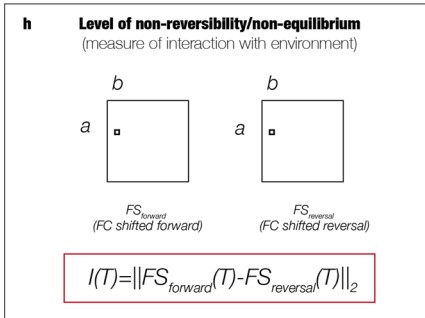

**Fig. 1 Capturing signatures of brain states estimating the inside out balance of intrinsic and extrinsic dynamics.** The arrow of time in brain signals contains a precise signature of the reversibility, reflecting how the brain is driven by the level of interaction with the environment. **a** In thermodynamics, the arrow of time in physical systems can be precisely estimated to provide a measure of non-reversibility and non-equilibrium, i.e. how the system is driven by external forces. The subpanel shows an example of a non-equilibrium system given by a Brownian particle in a moving potential, whose position is controlled externally, moving from one position to another in the forward process. The top of the subpanel shows examples the forward evolving trajectories (light grey lines) with the average (solid grey line). The average trajectory lags behind the centre of the potential (grey dashed line). In contrast (bottom of right subpanel) shows sample backward trajectories (light red lines) and their average (solid red line). Here the average trajectory leads the potential's centre (red dashed line). This asymmetry in time of forward and backward trajectories (i.e. the differences between the lines for average and the potential's centre) provides the exact level of non-reversibility/non-equilibrium of the system [adapted from[18]]. **b** The present framework uses this key idea from thermodynamics to extract the arrow of time in brain signals in order to capture the level of interaction between the brain and the environment. **c** The framework estimates the asymmetry by using pairwise comparisons of forward signals across the whole brain. **d** This is accomplished by constructing the backward brain signal from the forward signal by creating a reversal of the backward time series in each brain region. The panel shows the notation used for describing this process for a pair of regions. **e** The panel shows the main principle of the framework for measuring the level of non-reversibility/non-equilibrium through the pairwise level of asymmetry using a time-shifted measure of their correlation. The subpanels show how the shifted correlation captures the causal interactions between two time series where the top example shows strong time dependency while the bottom example shows weak time dependency. This is clearly seen by how the shifted correlation (as a function of the time shift, Δt) decays more rapidly for signals with weak compared to strong time dependency. The middle subpanels show the same method but now used for comparing forward and reversed regionals pairs of brain signals. The right subpanel shows examples of the time series for a given shift Δt = T and how the level of non-reversibility is computed as the absolute quadratic difference between the time-shifted correlations between forward and the reversal time series averaged over all pairs. **f** We applied this framework to the multidimensional time series covering the whole brain. **g** We created two time-shifted correlation matrices for the forward and reversed time series (at a given shift time point Δt = T). **h** The level of non-reversibility/non-equilibrium is given exactly by the distance between the two matrices (see Methods).

reversed trajectories provide the exact level of non-reversibility/non-equilibrium of the system. For a system in equilibrium with non-existent external forces moving the potential, the differences between the average and the potential's centre are close to zero for both forward and reversed cases. On the other hand, when an external force is moving the potential, this difference grows in opposite ways, where the difference in the forward case becomes more positive and more negative for the difference in the reversed case. This asymmetry in time reflects the arrow of time, indicating that the system moves further and further away from equilibrium into non-equilibrium.

This general principle can be directly applied to brain signals in order to capture the level of how much the brain is being driven by external forces from the environment. The construction of a reversed timeseries is done by reversing in time the natural forward evolution of the timeseries (Fig. 1b). The causal comparison of these two versions of a timeseries provides the foundation of the INSIDEOUT framework.

The general principle of how the estimation of the arrow of time between two timeseries is sketched in Fig. 1c–e. The precise notation used in the Methods for the construction the reversed brain signal from the forward signal is shown in Fig. 1d. The main principle of the INSIDEOUT framework for measuring the level of non-reversibility/non-equilibrium through the pairwise level of asymmetry using a time-shifted measure of their correlation (shown in Fig. 1e). The fundamental principle is described by providing two examples of where a timeseries (a) has a strong (top row) dependency on one timeseries (b) but a weak dependency on another timeseries (c, see insert). The framework measures this dependency through the time-shifted correlation between the timeseries. As can be seen, for the strong dependency, the slope of the decay is much flatter than for the weak dependency. Now, to determine the arrow of time, the framework computes the time-shifted correlation between the forward and the reversed timeseries (middle column). At a given shift ($\Delta t = T$, see line and arrow in figure), the level of non-reversibility can be computed as the absolute difference between the value of the forward and the reversed time-shifted correlations at this point in time.

Subsequently this is extended to multidimensional time series. We compute the pairwise time-shifted correlations (at a given shift time point $\Delta t = T$) which generates resulting time-shifted correlation matrices for the forward and reversed time series (shown in Fig. 1f–g). The comparison for extracting the level of non-reversibility/non-equilibrium is performed for all pairwise timeseries and therefore given exactly by the distance between the two matrices (see Methods).

The INSIDEOUT framework was applied to a unique dataset of four non-human primates in different brain states (wake, sleep and four different forms of anaesthesia), where the environment across conditions was the same. This allowed to precisely estimate a signature of a brain state characterised by the effect of the external, extrinsic environment on the intrinsic brain dynamics.

The experimental paradigm[29,30] included two major brain state manipulations: sleep and anaesthesia (Fig. 2a). The non-human primate was woken from sleep for the sleep condition (top row), leading to three periods of sleep, waking with eyes closed, and awake with eyes open. The anaesthesia protocol consisted of using one of four pharmacological agents (propofol, medetomidine (MD), ketamine+MD, and ketamine). For each anaesthetic condition, the row shows the conditions with (1) awake eyes closed, (2) awake eyes open, (3) anaesthetic, (4) recovery eyes closed, and (5) recovery eyes open. Across all sessions, the data was recorded with ECOG from 128 electrodes covering one hemisphere in each of the non-human primates. The combined

electrode locations across monkeys (Fig. 2b) as well as the locations in each individual shown in the other four figures. In order to minimise any potential confounds, we only used data from sessions with eyes closed. Similarly, to avoid potential volume conduction artefacts, the analysis used a PCA reduction of the dimensionality of the data (see Methods).

We hypothesised that the INSIDEOUT framework would be able to provide significantly different signatures for each of the five brain states. The results of analysing all the available (eyes closed) data from the non-human primates in sleep and four different states of anaesthesia are presented in Fig. 3. Importantly, as shown in the figure, conventional measures of functional connectivity fail to distinguish between these radically different brain states. The column shows individual examples of the functional connectivity matrices across the 128 electrodes for each condition and the scatter plots of the correlation between the conditions. As can clearly be seen from the scatter plots, this conventional method is unable to distinguish between the brain states, despite them being radically different. We performed the group level comparisons across all sessions and subjects, yielding a statistically non-significant result (Wilcoxon rank sum, $p > 0.05$). Equally, Supplementary Fig. 1 shows the scatterplots of the PC variances (diagonal elements of the functional connectivity in PCA space with $N = 10$ components) as the INSIDEOUT framework, which is also not useful for distinguishing brain states.

In contrast, as shown in Fig. 3b, c, the INSIDEOUT framework can clearly distinguish the brain states, using in all cases, a common optimal shift time point $T = 4$ for estimating the arrow of time. The middle panels of the figure show the level of non-reversibility for the same examples of individual sessions used in the functional connectivity analysis for a single subject. Computing this across all states, sessions and subjects, we found highly significant group level results, as shown in the violin plots with three stars indicating Wilcoxon rank sum ($p < 0.001$). As can be seen, the level of non-reversibility/non-equilibrium for sleep and all anaesthesia conditions (except for ketamine) is lower than in the awake state. This indicates that extrinsic environment is less important for driving intrinsic brain dynamics in states with less conscious awareness. Importantly, note that the level of non-reversibility/non-equilibrium increases again during the awake recovery session after anaesthesia (except ketamine) to similar levels as the initial awake session. The differential effects of ketamine compared to other anaesthetic agents are well known[31], arising from ketamine's dissociative effects[32,33]. Due to the fact that ketamine acts primarily as an antagonist of glutaminergic NMDA receptors, widespread and weak excitation appears in the brain, leading to complex, conscious experiences, including out-of-body experiences, and hallucinations[34]. As such it is a remarkable and a clear strength of the INSIDEOUT framework that it is able demonstrate how ketamine changes the non-reversibility/non-equilibrium differently from other anaesthetic agents.

In order to check whether it would make a difference to use a common PCA reference for all sessions in a given monkey, we compared this with our strategy of computing the PCA based on each session for each monkey in each condition. To compare these strategies, we concatenated all the sessions for one monkey (Chibi) in awake and in sleep and compared the reversibility on this common reference strategy compared to using PCAs of independent sessions in Supplementary Fig. 2. As can be seen, both strategies result in a similar level of significant differences in reversibility. Given that we cannot use a common reference for different monkeys and it is high-risk to concatenate sessions, we therefore used the independent PCA strategy as the more conservative choice.

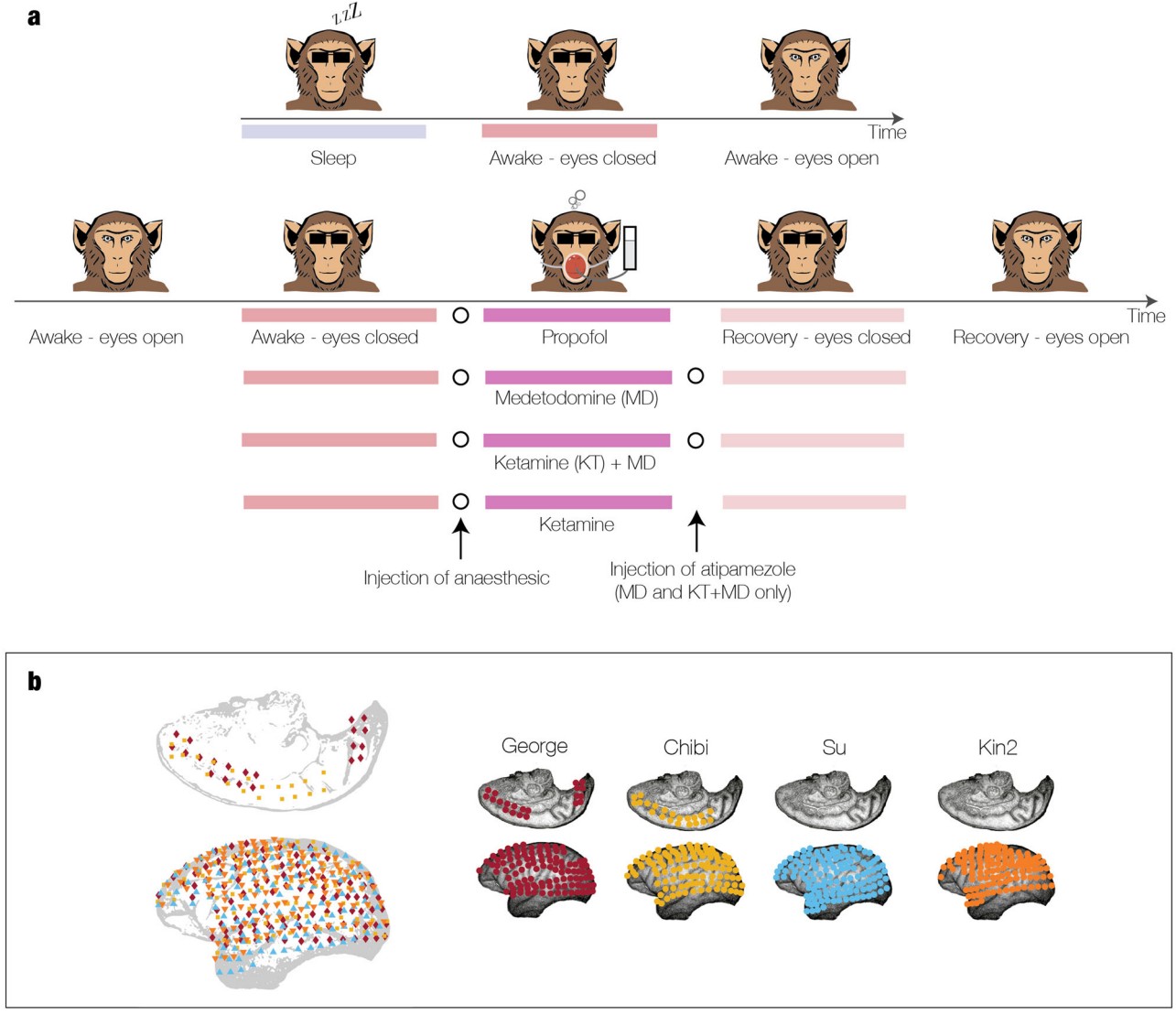

**Fig. 2 Experimental paradigm with different brain states in the same environment. a** We leveraged the power of an existing dataset from four non-human primates in different brain states (wake, sleep and four different forms of anaesthesia, see Methods for detailed description). The experimental paradigm consisted of two main manipulations of brain state measuring sleep and anaesthesia. For the sleep condition in the upper row, the non-human primate was awakened from sleep, leading to three sessions with sleep, awake with eyes closed and awake with eyes opened. For the anaesthesia conditions, the lower rows show the protocol with the injection of one of four pharmacological agents (propofol, medetomidine (MD), ketamine+MD and ketamine). The five sessions consisted of (1) awake eyes open, (2) awake eyes closed, (3) anaesthesia, (4) recovery eyes closed and (5) recovery eyes open. For the results reported here, only sessions with eyes closed were used. **b** The data from the sessions were recorded with electrocorticography (ECOG) from 128 electrodes covering one hemisphere in each of the non-human primates, with the combined locations shown in the subpanel and the locations in each individual shown in the other four subpanels.

Furthermore, we studied the influence of the number of PCA components as shown in Supplementary Fig. 3, where Supplementary Fig. 3a shows the variance of each PCA component for one monkey (Chibi) for all conditions, where they are clearly overlapping. For $N = 10$, we explain over 90% of the variance and we therefore chose this as the number of PCAs. Nevertheless, Supplementary Fig. 3b shows the level of non-reversibility for awake and sleep in the same monkey for all even numbered PCAs from [4..18]. Except for $N = 4$, all show a significant difference in non-reversibility between the two conditions. Finally, Supplementary Fig. 3c shows the p-values in Supplementary Fig. 3b.

We further investigated the validity and interpretation of the INSIDEOUT framework by computing the causal interactions between different regions using transfer entropy, which is an information-based measure of Granger causality. Using the NDTE framework[35], Supplementary Fig. 4 shows the results of comparing the five conditions in terms of comparing the levels of asymmetry, measured as the quadratic differences between the transfer entropy matrices (flow between pair of regions) and their transposed (see Methods). This level of asymmetry is a proxy for the breaking of the detailed balance. As can be seen from the figure, the NDTE results are consistent with those found with the INSIDEOUT framework and thus validates this. They also strengthen the interpretation of the link between non-reversibility/non-equilibrium and breaking the detailed balance.

In addition to the level of non-reversibility/non-equilibrium, we were interested in constructing a measure of hierarchy which could be used to estimate the level of orchestration changing according to the extrinsic driving of the environment.

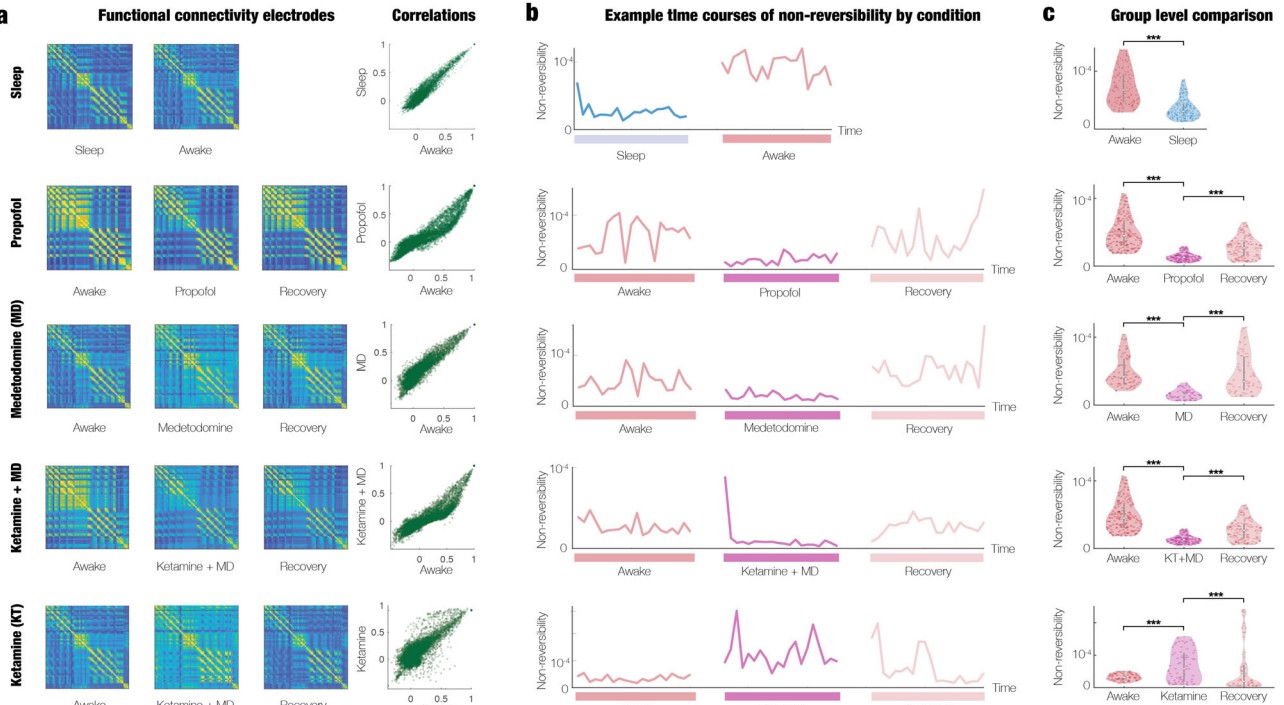

**Fig. 3 Estimating the arrow of time in brain signals provides significantly different signatures of brain state.** The figure summarises the results of analysing all the available (eyes closed) data from the non-human primates in sleep and four different states of anaesthesia (see Methods and Fig. 2). **a** The column shows individual examples of the functional connectivity matrices across the 128 electrodes and a scatter plot of the correlation between each condition. As can be seen, this conventional method is unable to distinguish between the very different brain states. **b** In contrast, the central panel shows the application of the thermodynamic framework of estimating the arrow of time on the same individual time series, which is clearly distinguishing brain states. **c** The column shows the significant group level results (across all non-human primates with all sessions belonging to the specific brain state) which provides a clear signature able to distinguish between different brain states (Wilcoxon rank-sum, $p < 0.001$).

This measure of hierarchy was computed as the variability of the non-equilibrium/non-reversibility of different brain states across space (see Methods). The five conditions are shown in Fig. 4a, where each row showing example renderings of the electrode-level non-equilibrium/non-reversibility for the two non-human primates participating in all experimental conditions. It is of considerable interest that the electrode-level renderings are not uniform but heterogenous across space in the different brain states. In general, there are more yellow electrode nodes (indicating high levels of non-reversibility) in awake than in sleep and anaesthesia. Furthermore, these regions are located near lower levels of the hierarchy in, for example, somatosensory regions. This strongly suggests that the hierarchy changes in each brain state as a function of how environment interacts with the intrinsic dynamics and thereby changing the important balance between intrinsic and extrinsic dynamics. Confirming this finding, the group-level results for the hierarchy of the 'inside out' balance of intrinsic and extrinsic balance across all sessions and all subjects shows highly significant signatures of the specific brain state (Wilcoxon rank-sum, $p < 0.001$). Interestingly, again there is a change in the opposite direction for ketamine.

This change in the hierarchy is driven by the breaking of the detailed balance associated with each condition. This is shown in the small inserts in Fig. 4, which render the detailed balance in terms of the causal driving of the incoming $FC_{in}(\tau)$ (top) and outgoing $FC_{out}(\tau)$ (bottom) information captured by the shifted correlation (see Fig. 1 and Methods). In all renderings, in both non-human primates in all conditions, a larger breaking of the balance is associated with a larger level of non-reversibility. As can be seen, in general anaesthesia the causal driving of the regions measured by the electrodes is more homogeneous than in

awake and recovery. Interestingly, this is not the case for sleep, which is almost as heterogeneous as the awake condition, perhaps attesting to the fact that sleep is naturally occurring brain state and not as disruptive as the pharmacological intervention used for anaesthesia.

Finally, the results reflect an interesting feature of the experimental setup, which is that the waking up from anaesthesia for the MD and KT + MD conditions required the injection of atipamezole (an antagonist to medetomidine). When comparing the awake state with the anaesthesia and recovery states for these two conditions (see Figs. 3 and 4), the group results show more comparable levels of non-reversibility and hierarchy between the recovery and the awake rather than the anaesthesia state. In contrast, for the propofol and ketamine conditions, the recovery condition is closer to the anaesthesia than the awake state (but still significantly different). This could indicate that the recovery was efficiently achieved when injecting atipamezole in the MD and KT + MD conditions.

In order to further validate the INSIDEOUT framework, we applied it to large-scale human HCP functional MRI neuroimaging data (see Methods). As shown in Supplementary Fig. 5, the results show that the level in non-reversibility increases in seven tasks (covering the full cognitive domain) compared to rest, consistent with other findings[21,23].

Further validating the INSIDEOUT framework with different brain states, Supplementary Fig. 6 shows significant differences between wakefulness and deep sleep brain states in human fMRI (see Methods). As such, these findings open up for future research, precisely characterising the arrow of time and non-reversibility/non-equilibrium in human brain states in health and disease.

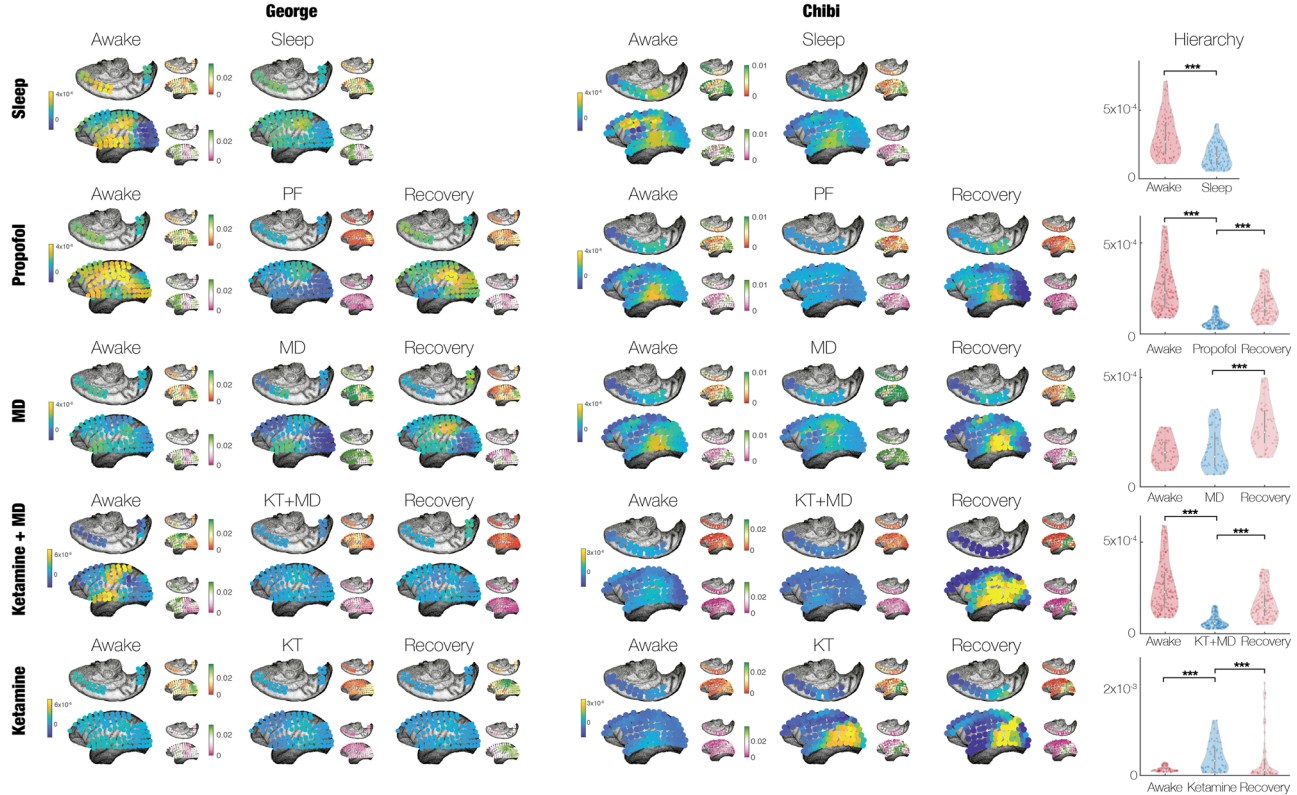

**Fig. 4 Hierarchical signature of different brain states.** We constructed a measure of hierarchy by estimating the variability of the non-equilibrium/non-reversibility of different brain states across space. For each condition, the row shows example renderings of the electrode-level non-equilibrium/non-reversibility for the two non-human primates (George and Chibi) who participated in all experimental conditions. The smaller inserts show how the level of reversibility is associated with the causal driving of the incoming $FC_{in}(\tau)$ and outgoing $FC_{out}(\tau)$ information captured by the shifted correlation (see Fig. 1 and Methods). The column shows the significant group-level results for the hierarchy of the inside out balance of intrinsic and extrinsic balance for all non-human primates with all sessions belonging to the specific brain state (Wilcoxon rank-sum, $p < 0.001$).

## Discussion

A current focus of intensive investigation in the field is how to extract dynamical signatures of distinct brain states. We addressed this challenging question by using the INSIDEOUT framework, which allowed us to discover how different brain states react to the same environment, giving rise to significantly different signatures in the balance of intrinsic and extrinsic brain dynamics in sleep and anaesthesia.

The key idea of the INSIDEOUT framework comes from thermodynamics, which uses the concept of arrow of time to characterise the level of the non-reversibility/non-equilibrium in brain signals. The asymmetry in temporal processing of the environment, i.e. the level of reversibility, directly measures the breaking of the detailed causal interactions found in the brain dynamics. In other words, our method is an alternative way to capture differences in effective connectivity and hierarchical processing in different brain states. Compared with other related methods for effective connectivity such as that of Gilson and colleagues[36], the INSIDEOUT framework does not need to assume any underlying model. In addition, the present framework is also much simpler compared to transfer entropy methods[35,37], given that it is based on time-shifted correlation. The simplicity of the framework also provides a way of estimating the arrow of time or reversibility which approximates the true measure of production entropy. This is important since it is very difficult to fully compute. To the best of our knowledge, there currently only two papers in the literature that estimates the production entropy[21,22]. The former uses the flow of the system while the latter use spin models to establish a link between the breaking of the detailed balance and reversibility. This establishes the exact mathematical relationship between intrinsic and extrinsic dynamics, but through an approximation in a low-dimensional space reduction. The advantage of the simpler INSIDEOUT framework is that it can capture the arrow of time across all time signals, whether in ECOG or in fMRI, as shown here.

Crucially, the INSIDEOUT framework was able to reveal significantly different signatures of non-reversibility/non-equilibrium and hierarchy in ECOG brain data from non-human primates in three radically different brain states (awake, deep sleep and anaesthesia), which are difficult to capture with conventional methods. This provided a quantification of the differential breaking of the detailed balance in different brain states, where a change in the detailed balance of the brain dynamics was defined as the change in the hierarchy of the causal interactions in the intrinsic brain dynamics, i.e. the breaking of the balance reflects the level of asymmetry of causal interactions, which is captured through the arrow of time. Specifically, we found that the level of non-reversibility/non-equilibrium was higher in wakefulness than both sleep and anaesthesia. This significant feature of non-wakeful states was also shown by a flatter hierarchy reflecting the diversity of the reversibility across the brain. In summary, this thermodynamics-inspired framework offers a quantification of signatures of the 'inside out' balance of intrinsic and extrinsic brain dynamics, which is closely linked to subjective conscious awareness.

One of the paradoxes of the brain is how fundamentally different brain states such as wakefulness, sleep or anaesthesia can

emerge from the same underlying fixed anatomy. Over longer periods of time, the brain can exhibit neuroplasticity as part of learning, but it has remained a challenging question that even without neuroplasticity different brain states can coexist in the same static anatomy of the so-called connectome.

Exciting research has proposed that different brain states are characterised by different dynamical regimes[2,4,38]. Indeed, it was shown that both at macroscopic and microscopic scales, unconscious brain states are dominated by synchronous activity[2,38–41], while conscious states are characterised by asynchronous dynamics[38,42,43]. Equally, in non-human primates, research has investigated changes in anaesthesia, where for example maximum entropy models were used to derive collective, macroscopic properties that quantify the dynamics in a brain state that produce work, contain and transmit information[44]. Another way to quantify different brain state dynamics is to measure the hierarchy using a method called intrinsic ignition, which determines the internal propagation of signals across the brain[45,46]. This was used to analyse resting-state fMRI data in awake and anesthetized non-human primates to reveal spatial and temporal hierarchical differences between the brain states[47]. Using whole-brain modelling of such profound dynamical changes between brain states revealed changes in global synchronisation and other metrics such as functional connectivity, structure-function relationship, integration and segregation across vigilance states[48].

In humans, neuroimaging studies have also investigated changes in many brain states including wakefulness, meditation, anaesthesia, sleep and even coma[3,49,50]. A comprehensive study of brain states showed significant different levels of turbulent dynamics[3]. Whole-brain modelling of wakefulness, anaesthesia, and coma found reduced network interactions, together with more homogeneous and more structurally constrained local dynamics for brain states with reduced levels of conscious access[49]. Furthermore, the non-stationary landscapes of different brain states were described by a mathematical framework that characterised differences in the attractor structure, i.e. the stationary points and their connections over time[51].

Ultimately, however, the paradox of how brain states can coexist in the same static connectome can perhaps best be solved by considering the changes in intrinsic properties of the connectome (such as neuromodulators[4,52,53]) and extrinsic, driving forces from the environment. Therefore, a likely signature of the complex dynamics of self-organised activity in a given brain state must arise from a description of the balance between intrinsic and extrinsic dynamics. This is exactly the hypothesis driving the INSIDEOUT framework described here. Previous research has elegantly described that the brain activity following extrinsic perturbation depends on the intrinsic dynamics and that this is different in different brain states, including wakefulness, sleep, anaesthesia, and post-coma states[11–15]. In contrast, the INSIDEOUT framework proposed here is able to quantify how the same natural extrinsic environment can lead to a radical different balance between intrinsic and extrinsic dynamics for a given brain state.

Furthermore, the results quantify the concepts of prominent theories of consciousness such as the Global Workspace[35,54,55], Integrated Information Theory[6] and the Temporo-spatial Theory of Consciousness[7] in terms of the arrow of time. Here we show that compared to non-conscious brain states, conscious awareness is associated with greater non-reversibility/non-equilibrium reflecting a richer hierarchy associated with more breaking of the detailed balance of the causal interactions in brain dynamics.

A fundamental question in general biology is how life depends on survival and how this requires the ability to find order in a complex, largely disordered environment. Survival can be described as the avoidance of decay and of equilibrium, as

proposed by the Erwin Schrödinger[20]. The brain controls these non-equilibrium interactions between internal brain dynamics and the complex environment through the regulation of the 'inside out' balance of intrinsic and extrinsic dynamics.

Previous research in thermodynamics provides a convenient framework for describing how a non-equilibrium system––where detailed balance is broken––shows net fluxes between the underlying states, and thus becomes irreversible, establishing an arrow of time[18,24–27]. These non-equilibrium dynamics can be described by turbulence, which is highly useful for optimally transferring energy/information over spacetime due to its mixing properties[56].

The key idea behind the INSIDEOUT framework is to quantify how the environment drives the internal brain dynamics by measuring the non-reversibility/non-equilibrium in brain signals through the arrow of time. Previous research has measured the entropy production to characterise non-equilibrium[21,22]. Yet, other methods have directly estimated the arrow of time by using deep learning algorithms to capture the non-equilibrium of brain states[23,57]. In contrast, here we use a much faster and more sensitive method of estimating the asymmetry in time through using the multivariate pairwise shifted correlation of the forward and reversed timeseries. The results show that this approach provides clear signatures of the different brain states. It is of interest to note that the optimal time shifting parameter ($T = 4$) establishes a timescale for these signatures. Other important research on timescales has shown a shift towards slower frequencies in brain states with sensory deficits (such as sleep, unresponsive wakefulness syndrome and anaesthesia) rather than motor deficits (such as locked-in syndrome)[58].

Interestingly, the link between non-equilibrium dynamics and turbulence, which are found in many physical systems is also found in the human brain, where turbulence and the resulting information cascade play a key role extracting order from disorder in the environment[59–62]. Turbulence in the human brain is poised on the edge of criticality, between fluctuations to oscillations[63], which creates a straightforward signature for describing the different degrees of dynamical flexibility in brain states[3].

The INSIDEOUT framework provided precise signatures for the brain states of wakefulness, deep sleep and anaesthesia in an ECOG dataset acquired in non-human primates. The signatures for each brain state were different in terms of the reversibility and hierarchy of their underlying large-scale dynamics. The arrow of time capturing the level of reversibility measures the influence of the asymmetry in temporal processing of the environment on the intrinsic brain dynamics. The breaking of the detailed causal interactions found in brain dynamics is lower in both deep sleep and anaesthesia than in wakefulness. This is also reflected in a flatter hierarchy, measured as the variability of the level of non-reversibility/non-equilibrium across the brain. This means that when the level of subjective vigilance under anaesthesia and sleep is reduced, the same extrinsic environment is driving the intrinsic dynamics to a significant lower degree. This 'inside out' balance is reflected in the level of reversibility and therefore sufficient to fully distinguish between different brain states and wakefulness.

Furthermore, the INSIDEOUT framework is sufficiently sensitive to detect the differential effects of ketamine compared to other anaesthetic agents[31], perhaps reflecting what is known as "dissociative anaesthesia"[32,33]. Ketamine is known to cause widespread, weak excitation in the brain, given that it acts primarily as an antagonist of glutaminergic NMDA receptors[34].

Another remarkable differentiation is how the group results show more comparable levels of non-reversibility and hierarchy between the recovery and the awake condition rather than the anaesthesia state for the medetomidine, possibly due to the fact

that the waking up from anaesthesia for the MD and KT + MD conditions required the injection of atipamezole (the antagonist to medetomidine). This was not the case for the propofol and ketamine conditions, where the recovery condition was closer to anaesthesia than the awake state. In the case of propofol, one of the most commonly used anaesthetics in medicine, the mechanism of action is not fully understood but clearly linked to widespread modulation of GABAA receptors[64]. Through the binding to GABAA receptors, the effects of endogenous GABA allow propofol to cause widespread inhibition of neuronal activity, which at higher doses lead to full anaesthesia and, at lower doses, states of sedation, amnesia and atonia. On the other hand, as mentioned above, ketamine is an antagonist of glutamateric NMDA receptors with local excitatory response leading to widespread, weak stimulation of brain activity.

Overall, we have demonstrated the ability of the INSIDEOUT framework to extract differential signatures of fundamentally different brain states in the fixed anatomical connectome through measuring the 'inside out' balance of intrinsic and extrinsic dynamical interactions. This framework carries within the echoes of the writings of the medieval scholar Thomas Aquinas who presciently wrote 'cognoscentia a non cognoscentibus in hoc distinguuntur, quia non cognoscentia nihil habent nisi formam suam tantum; sed cognoscens natum est habere formam etiam rei alterius, nam species cogniti est in cognoscente'[65]. This roughly translates into '… the cognizant are distinguished from the non-cognizant in this respect, that the non-cognizant have nothing but their own form alone, whereas a cognizant entity is disposed to have the form of another thing as well. For the species of the thing known is in the knower'. This stresses precisely the difference between the content and the container, in cognitive terms between the mind and the environment and in physical terms between the intrinsic and the extrinsic dynamics. Perhaps in future, taking inspiration from recent work on the frontiers of science[66], the INSIDEOUT framework could potentially serve as an individual 'fingerprint' unique to each individual brain, and thus test the implicit idea in Aquinas that individual cognizants might differ between them.

To sum up, here we provide precise signatures of distinct, different brain states by reformulating a central question in neuroscience, namely how intrinsic dynamics in a brain state are differentially shaped by the same extrinsic environment. This allows us to determine the balance between intrinsic and extrinsic dynamics by establishing the temporal asymmetry of large-scale brain signals through an estimation of the arrow of time. As such this reveals the hierarchical orchestration of ongoing dynamics which is important for promoting survival, with sensory regions lower in the hierarchy more extrinsically driven by the environment, while regions in the top of hierarchy are more intrinsically and less extrinsically driven.

## Methods
### ECOG data
*Neuroimaging Ethics.* The RIKEN ethical committee (No. H24-2-203(4)) approved all experimental and surgical procedures and the research carefully followed the recommendations of the Weatherall report "The use of non-human primates in research".

*Experimental setup.* As described in details in the original paper[30], overall care was managed by the Division of Research Resource Center at RIKEN Brain Science Institute, where the non-human primates were housed in a large individual enclosure with other animals visible in the room, and maintained on a 12:12-h light:dark cycle. The multidimensional recording technique used chronically implanted, customized multichannel ECoG electrode arrays (Unique Medical, Japan). The 128 electrodes array (made of 3-mm diameter platinum discs), was implanted in the subdural space in 4 adult macaque monkeys (George, Chibi, Su and Kin2). The array had an interelectrode distance of 5 mm and were implanted

in the left hemisphere, continuously covering over the frontal, parietal, temporal, and occipital lobes (Fig. 2b, the detailed methods are described in[29]).

*Data.* Electrocorticography (ECoG) recordings were obtained in non-human primates in different brain states (awake and deep sleep as well as anaesthetic doses of propofol, MD, KT + MD and KT. See further details in[29,30]. Recordings were obtained from 128 channels, with an example layout shown in Fig. 2b. All the sessions used here were conducted with eyes closed. For more information on the dataset visit http://neurotycho.org/.

*ECoG data pre-processing.* The pre-processing consisted of applying a notch filter to eliminate line noise and its harmonics (50 Hz, 100 Hz, 150 Hz); the timeseries were bandpass filtered between 5 and 500 Hz, and resampled from 1 KHz to 256 Hz and finally z-scored, similar to the procedure described in[22,57]. Across all subjects and sessions (eyes closed) this yielded a total of 3522 s of data for wakefulness and 26660 s for sleep. For the three sessions of ketamine (wakefulness, anaesthesia and recovery) there were 3353, 2210 and 3063 s of data, respectively. For KT + MD, there were 11603, 18244 and 22371 s of data; for propofol: 3687, 2324 and 3603 s of data; for MD: 3438, 3160 and 2791 s of data. For subject, the timeseries for each session was then inverted in time, giving a forward and a reversed sample timeseries which was used for the analysis.

### Human Connectome project: Acquisition and pre-processing
*Ethics.* The Washington University–University of Minnesota (WU-Minn HCP) Consortium obtained full informed consent from all participants, and research procedures and ethical guidelines were followed in accordance with Washington University institutional review board approval (Mapping the Human Connectome: Structure, Function, and Heritability; IRB # 201204036).

*Participants.* The data set used for this investigation was selected from the March 2017 public data release from the Human Connectome Project (HCP) where we chose a sample of 1003 participants, all of whom have resting state data. For the seven tasks, HCP provides the following numbers of participants: WM = 999; SOCIAL = 996; MOTOR = 996; LANGUAGE = 997; GAMBLING = 1000; EMOTION = 992; RELATIONAL = 989. No statistical methods were used to predetermine sample sizes but our sample sizes are similar to those reported in previous publications using the full HCP dataset.

*The HCP task battery of seven tasks.* The HCP task battery consists of seven tasks: working memory, motor, gambling, language, social, emotional, relational, which are described in details on the HCP website[67]. HCP states that the tasks were designed to cover a broad range of human cognitive abilities in seven major domains that sample the diversity of neural systems (1) visual, motion, somato-sensory, and motor systems, (2) working memory, decision-making and cognitive control systems; (3) category-specific representations; (4) language processing; (5) relational processing; (6) social cognition; and (7) emotion processing. In addition to resting state scans, all 1003 HCP participants performed all tasks in two separate sessions (first session: working memory, gambling and motor; second session: language, social cognition, relational processing and emotion processing).

*Neuroimaging acquisition for fMRI HCP.* The 1003 HCP participants were scanned on a 3-T connectome-Skyra scanner (Siemens). We used one resting state fMRI acquisition of approximately 15 min acquired on the same day, with eyes open with relaxed fixation on a projected bright cross-hair on a dark background as well as data from the seven tasks. The HCP website (http://www.humanconnectome.org/) provides the full details of participants, the acquisition protocol and pre-processing of the data for both resting state and the seven tasks. Below we have briefly summarised these.

The pre-processing of the HCP resting state and task datasets is described in details on the HCP website. Briefly, the data is pre-processed using the HCP pipeline which is using standardized methods using FSL (FMRIB Software Library), FreeSurfer, and the Connectome Workbench software[68,69]. This standard pre-processing included correction for spatial and gradient distortions and head motion, intensity normalization and bias field removal, registration to the T1 weighted structural image, transformation to the 2 mm Montreal Neurological Institute (MNI) space, and using the FIX artefact removal procedure[69,70]. The head motion parameters were regressed out and structured artefacts were removed by ICA + FIX processing (Independent Component Analysis followed by FMRIB's ICA-based X-noiseifier[71,72]). Pre-processed timeseries of all grayordinates are in HCP CIFTI grayordinates standard space and available in the surface-based CIFTI file for each participants for resting state and each of the seven tasks.

We used a custom-made Matlab script using the ft_read_cifti function (Fieldtrip toolbox[73]) to extract the average timeseries of all the grayordinates in each region of the Mindboggle-modified Desikan-Killiany parcellation[74] with a total of 62 cortical regions (31 regions per hemisphere)[75], which are defined in the HCP CIFTI grayordinates standard space. The BOLD timeseries were filtered using a second-order Butterworth filter in the range of 0.008–0.08 Hz.

## Human sleep data: Acquisition and pre-processing

*Ethics.* Written informed consent was obtained, and the study was approved by the ethics committee of the Faculty of Medicine at the Goethe University of Frankfurt, Germany.

*Participants.* We used fMRI- and PSG data from 18 participants taken from a larger database that reached all four stages of PSG[76,77]. Exclusion criteria focussed on the quality of the concomitant acquisition of EEG, EMG, fMRI, and physiological recordings.

*Acquisition and pre-processing of fMRI and polysomnography data.* Neuroimaging fMRI was acquired on a 3 T system (Siemens Trio, Erlangen, Germany) with the following settings: 1505 volumes of T2*-weighted echo planar images with a repetition time (TR) of 2.08 s, and an echo time of 30 ms; matrix 64 × 64, voxel size $3 \times 3 \times 2$ mm³, distance factor 50%, FOV 192 mm².

The EPI data were realigned, normalised to MNI space, and spatially smoothed using a Gaussian kernel of 8 mm³ FWHM in SPM8 (http://www.fil.ion.ucl.ac.uk/spm/). Spatial downsampling was then performed to a $4 \times 4 \times 4$ mm resolution. From the simultaneously recorded ECG and respiration, cardiac- and respiratory-induced noise components were estimated using the RETROICOR method[78], and together with motion parameters these were regressed out of the signals. The data were temporally bandpass filtered in the range 0.01–0.1 Hz using a sixth-order Butterworth filter. We extracted the timeseries in the AAL parcellation[79].

Simultaneous PSG was performed with the recording of EEG, EMG, ECG, EOG, pulse oximetry, and respiration. EEG was recorded using a cap (modified BrainCapMR, Easycap, Herrsching, Germany) with 30 channels, of which the FCz electrode was used as reference. The sampling rate of the EEG was 5 kHz, and a low-pass filter was applied at 250 Hz. MRI and pulse artefact correction were applied based on the average artefact subtraction method[80] in Vision Analyzer2 (Brain Products, Germany). EMG was collected with chin and tibial derivations, and as the ECG and EOG recorded bipolarly at a sampling rate of 5 kHz with a low-pass filter at 1 kHz. Pulse oximetry was collected using the Trio scanner, and respiration with MR-compatible devices (BrainAmp MR+, BrainAmp ExG; Brain Products, Gilching, Germany).

Participants were instructed to lie still in the scanner with their eyes closed and relax. Sleep classification was performed by a sleep expert based on the EEG recordings in accordance with the AASM criteria (2007). Results using the same data and the same pre-processing has previously been reported[76,77].

**INSIDEOUT framework and associated methods.** The INSIDEOUT framework is a general method which can use many types of data as specified in the following, where we use it on ECOG and neuroimaging fMRI data.

*Functional connectivity.* The functional connectivity in Fig. 3a are matrices of correlations across of time series activity of the 128 ECOG electrodes in the different brain states.

*Method for determining levels of non-reversibility/non-equilibrium.* Capturing the level of non-reversibility, and consequently the level of non-equilibrium forced by the driving of the external environment in the intrinsic dynamics, relies on the key idea of detecting the arrow of time through the degree of *asymmetry* obtained by comparing the causal relationship between pairwise time series of the forward and the artificially generated reversed backward version. More specifically, let's consider first the detection of the level of non-reversibility (i.e. the arrow of time) between two time series $x(t)$ and $y(t)$. As shown in Fig. 1d, let's assume that $x(t)$ is evolving from an initial state $A_1$ to a final state $A_2$, and $y(t)$ is evolving from an initial state $B_1$ to a final state $B_2$, respectively. The reversed backward version of $x(t)$ (or $y(t)$), that we call $x^{(r)}(t)$ (or $y^{(r)}(t)$), is obtained by flipping the time ordering, i.e. by ordering the time evolution of $x^{(r)}(t)$ (or $y^{(r)}(t)$) as the inverted sequence determined by initial state $A_2$ to a final state $A_1$ (or initial state $B_2$ to a final state $B_1$). The causal dependency between the time series $x(t)$ and $y(t)$ are measured through the time-shifted correlation. For the forward evolution the time-shifted correlation is given by

$$c_{forward}(\triangle t) = <x(t), y(t + \triangle t)> \tag{1}$$

and for the reversed backward evolution the time-shifted correlation is given by

$$c_{reversal}(\triangle t) = <x^{(r)}(t), y^{(r)}(t + \triangle t)> \tag{2}$$

The pairwise level of non-reversibility, i.e. the degree of temporal asymmetry capturing the arrow of time, is given consequently by the absolute difference between the causal relationship between these two timeseries in the forward and reversed backward evolution, at a given shift $\triangle t = T$, i.e.,

$$I_{x,y}(T) = \left| c_{forward}(T) - c_{reversal}(T) \right| \tag{3}$$

We selected the optimal $T = 4$ in a two-step procedure, where we first compute the averaged autocorrelation over all signals and identified the approximate value of T, where the autocorrelation has sufficiently decayed. We then optimise around this value to find the most significant results.

The level of non-reversibility/non-equilibrium for the multidimensional case can be easily generalized by defining the forward and reversal matrices of time-shifted correlations. Let's denote with $x_i(t)$ the forward version of a multidimensional time series reflecting the dynamical evolution of the variable describing the system. In this case the sub-index $i$ denotes the different dimensions of the dynamical system. Let's denote with $x_i^{(r)}(t)$ the corresponding reversed backward version. The forward and reversal matrices expressing the functional causal dependencies between the different variables for the forward and artificially generated reversed backward version of a multidimensional system are given by

$$FS_{forward,ij}(\triangle t) = -\frac{1}{2}\log\left(1 - <x_i(t), x_j(t + \triangle t)>^2\right) \tag{4}$$

$$FS_{reversal,ij}(\triangle t) = -\frac{1}{2}\log\left(1 - <x_i^{(r)}(t), x_j^{(r)}(t + \triangle t)>^2\right) \tag{5}$$

respectively. The $FS$ functional causal dependencies matrices are expressed as the mutual information based on the respective time-shifted correlations. The level of non-reversibility is given by the quadratic distance between the forward and reversal time-shifted matrices, at a given shift $\triangle t = T$. In other words, the level of non-reversibility/non-equilibrium in the multidimensional case is given by

$$I = ||FS_{forward}(T) - FS_{reversal}(T)||_2 \tag{6}$$

where the notation $||Q||_2$ is defined as the mean value of the absolute squares of the elements of the matrix Q. In other words, if we define a difference matrix $FS_{diff}$ in the following way

$$FS_{diff}, ij = (FS_{forward}, ij(T) - FS_{reversal}, ij(T))^2 \tag{7}$$

The matrix $FS_{diff}$ is thus a matrix whose elements are the squared of the elements of the matrix $(FS_{forward}(T) - FS_{reversal}(T))$, where for each pair, the level of non-reversibility as measured by the squared difference. Thus, $I$ is simply the mean value of the elements of $FS_{diff}$.

The hierarchy measure used in Fig. 4 is computed as the standard deviation of the elements of the matrix $FS_{diff}$, which is the matrix of the squared difference between forward and reversed time-shifted correlations. In other words, this measure of spatiotemporal hierarchy reflects the degree of asymmetries in the causal interactions between elements.

Further, to avoid volume conduction artifacts, we reduce the original system expressed by the $N = 128$ electrodes, denoted here by $\bar{X} = (X_1, \dots, X_N)$, to an $n$ dimensional system $x_i(t)$, given by the first $n = 10$ principal components which were able to capture over 95% of the variance of the empirical data in all subjects and sessions.

Mathematically, if we define a $M \times N$ matrix of the electrode data $D\prime$ where each of the $M$ rows represents a time point $t$ of the electrodes measurements, i.e. an instantiation of the vector $\bar{X} = (X_1, \dots, X_N)$. Let's be $D$ the matrix with column-wise zero empirical mean, i.e. where the sample mean of each column has been shifted to zero. Thus, in matrix form, the empirical covariance matrix for the original variables can be written as

$$Q = D^T D = W \Lambda W^T \tag{8}$$

Where $\Lambda$ is a diagonal matrix with ordered eigenvalues of the covariance matrix $Q$ and $W$ is a matrix with columns given by the corresponding eigenvectors. The reduced PCA system where the analysis is performed, i.e. $x_i(t)$ are the rows of the reduced matrix $T$

$$T = DW_n \tag{9}$$

Being $W_n$ the matrix built up by keeping the first $n$ columns, i.e. the first $n$ eigenvectors of the matrix $W$.

*Method for back projecting PCA space to electrode space.* In order to project back in the original $N$-electrodes space, the level of non-reversibility computed in the reduced $n$-dimensional PCA space, we computed a vector $J$ in PCA space reflecting in each component the averaged level of non-reversibility as following

$$J_i = \left(\frac{1}{n}\sum_{j=1}^{n} FS_{diff,ji} + \frac{1}{n}\sum_{j=1}^{n} FS_{diff,ij}\right)/2 \tag{10}$$

We averaged the components of $J$ for each non-human primate and each condition over all the corresponding measured sessions. We render the vector $J$ projected back in a vector $R$ in the $N$-electrode space by computing

$$R = JW_n^T \tag{11}$$

*The breaking of the detailed balance.* The breaking of the detailed balance can be captured by using the $FS_{forward}$ matrix. The incoming flow is defined as $FC_{in}(i) = \sum_j FS_{forward,ij}(T)$, while the outcoming flow is defined as $FC_{out}(i) = \sum_j FS_{forward,ji}(T)$.

**Normalised directed transfer entropy (NDTE).** We used our established NDTE framework[35] to compute the directed flow between brain regions. Briefly, this allows us to characterise the causal interaction between two brain regions by an information theoretical statistical criterion that allows us to infer the underlying

bidirectional reciprocal communication. The NDTE framework was inspired by the work of Brovelli and colleagues who used and validated a similar transfer entropy framework in neuroimaging data[37]. This framework uses a Gaussian approximation, i.e. only second-order statistics of the involved entropies, which means, as shown below, that instead of estimating the probabilities, the method estimates the covariance which massively facilitates computation.

Let us assume that we want to describe the statistical causal interaction exerted from a source brain area $X$ to another target brain area $Y$. We aim to measure the extra knowledge that the dynamical functional activity of the past of $X$ contribute to the prediction of the future of $Y$, by the following mutual information:

$$I(Y_{i+1}; X^i | Y^i) = H(Y_{i+1} | Y^i) - H(Y_{i+1} | X^i, Y^i) \quad (12)$$

where $Y_{i+1}$ is the activity level of brain area $Y$ at the time point $i+1$, and $X^i$ indicates the whole activity level of the past of $X$ (filtered ECOG signal) in a time window of length $T$ up to and including the time point $i$ (i.e. $X^i = [X_i X_{i-1} \ldots X_{i-(T-1)}]$). The time lag between $i$ and $i+1$ is $t_{lag}$. Note that this causality measure is not symmetric, i.e. allows bidirectional analysis. The conditional entropies are defined as follows:

$$H(Y_{i+1} | Y^i) = H(Y_{i+1}, Y^i) - H(Y^i) = -\sum_{y_{i+1}, y^i} p(y_{i+1}, y^i) \log(p(y_{i+1} | y^i)) \quad (13)$$

$$\begin{aligned} H(Y_{i+1} | X^i, Y^i) &= H(Y_{i+1}, Y^i, X^i) - H(X^i, Y^i) \\ &= -\sum_{y_{i+1}, y^i, x^i} p(y_{i+1}, y^i, x^i) \log(p(y_{i+1} | x^i, y^i)) \end{aligned} \quad (14)$$

The mutual information $I(Y_{i+1}; X^i | Y^i)$ expresses the degree of statistical dependency between the past of $X$ and the future of $Y$. In other words, if that mutual information is equal to zero, then the probability $p(Y_{i+1}, X^i | Y^i) = p(Y_{i+1} | Y^i) \cdot p(X^i | Y^i)$ and thus we can say that there is no causal interaction from $X$ to $Y$.

Consequently $I(Y_{i+1}; X^i | Y^i)$ expresses a strong form of Granger causality[81], by comparing the uncertainty in $Y_{i+1}$ when using knowledge of only its own past $Y^i$ or the past of both brain regions, i.e. $X^i, Y^i$. This information-theoretical concept of causality was introduced in neuroscience by Schreiber[82] and is usually called Transfer Entropy[37,83–85]. In order to facilitate computation, Brovelli et al.[37] proposed a weaker form of causality allowing calculation of the involved entropies by just considering a Gaussian approximation, i.e by considering only second-order statistics. Indeed, under this approximation, the entropies can be computed as follows:

$$H(Y^i) = \frac{T}{2} \log(2\pi e) + \frac{1}{2} \log(\det(\Sigma(Y^i))) \quad (15)$$

$$H(Y_{i+1}, Y^i) = \frac{T+1}{2} \log(2\pi e) + \frac{1}{2} \log(\det(\Sigma(Y_{i+1}, Y^i))) \quad (16)$$

$$H(X^i, Y^i) = T \log(2\pi e) + \frac{1}{2} \log(\det(\Sigma(X^i, Y^i))) \quad (17)$$

$$H(Y_{i+1}, Y^i, X^i) = \frac{2T+1}{2} \log(2\pi e) + \frac{1}{2} \log(\det(\Sigma(Y_{i+1}, Y^i, X^i))) \quad (18)$$

In other words, causality is based only on the corresponding covariance matrices.

In order to be able to sum and compare the directed mutual information flow between different pairs of brain regions, this has to be appropriately normalised. In fact, if the mutual information directed flow is correctly normalized then the different values could be combined for example to know the total directed flow exerted by the whole brain on a single region or vice versa, the directed flow exerted by a single brain region on the whole brain. More specifically, we define this information theoretical measure as normalised directed transfer entropy (NDTE) flow, $F_{XY}$, from timeseries $X$ to $Y$:

$$F_{XY} = I(Y_{i+1}; X^i | Y^i) / I(Y_{i+1}; X^i, Y^i) \quad (19)$$

where $I(Y_{i+1}; X^i, Y^i)$ is the mutual information that the past of both signals together, $X^i, Y^i$, has about the future of the target brain region $Y_{i+1}$. Given that,

$$I(Y_{i+1}; X^i, Y^i) = I(Y_{i+1}; Y^i) + I(Y_{i+1}; X^i | Y^i) \quad (20)$$

this normalisation compares the original mutual information directed flow, i.e. the predictability of $Y_{i+1}$ by the past of $X^i | Y^i$ with the internal predictability of $Y_{i+1}$, i.e. $I(Y_{i+1}; Y^i)$. We define the matrix of NDTE flow in the brain for a given number of N electrodes, $\mathcal{C}$ with elements: $\mathcal{C}_{ij} = F_{X(i)X(j)}$, where $X(k)$ corresponds to the filtered ECOG time series from electrode $k$. Here we used a time lag $t_{lag} = 4$ and embedding of the past of $T = 8$.

**Statistics and reproducibility.** All statistical analyses of the data conducted here used the standard statistical Wilcoxon rank-sum method (as specified in the text). Equally, all data is in the public domain (as stated in the Data availability) and the reproducibility is specified for each dataset in the original empirical papers.

**Reporting summary.** Further information on research design is available in the Nature Research Reporting Summary linked to this article.

## Data availability

The ECOG data is freely available from neurotycho.org[30]. The multimodal neuroimaging data are freely available from HCP[86]. The sleep data was provided by Dr Helmut Laufs[76]. In line with the journal policies, the datapoints for the violinplots in Figs. 3C and 4 are available in the supplementary data file SuppData1.xls.

## Code availability

The code used to run the analysis is available on GitHub (https://github.com/decolab/insideout), DOI: 10.5281/zenodo.6561284.

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

## Acknowledgements

G.D. was supported by the Human Brain Project Specific Grant Agreement 3 Grant agreement no. 945539 and by the Spanish Research Project AWAKENING: using whole-brain models perturbational approaches for predicting external stimulation to force transitions between different brain states, ref. PID2019-105772GB-I00/AEI/10.13039/501100011033, financed by the Spanish Ministry of Science, Innovation and Universities (MCIU), State Research Agency (AEI). Y.S.P is supported by European Union's Horizon 2020 research and innovation programme under the Marie Sklodowska-Curie grant 896354. E.T. is supported by grants PICT-2018-03103 and PICT-2019-02294 funded by Agencia I+D+I (Argentina) and by a Mercator fellowship granted by the German Research Foundation. M.L.K. is supported by the Center for Music in the Brain, funded by the Danish National Research Foundation (DNRF117), and Centre for Eudaimonia and Human Flourishing at Linacre College funded by the Pettit and Carlsberg Foundations.

## Author contributions

G.D., Y.S.P. and M.L.K. designed the study, developed the methods, performed the analyses, and wrote and edited the manuscript. E.T. & H.B. pre-processed the non-human primate data. All the authors edited the manuscript.

## Competing interests

The authors declare to have no conflict of interest. The funders had no role in study design, data collection and analysis, decision to publish or preparation of the manuscript.

E.T. is an Editorial Board Member for Communications Biology, but was not involved in the editorial review of, nor the decision to publish this article.
