## [Peer Review File · Communications Biology]

Reviewers' comments:

Reviewer #1 (Remarks to the Author):

This is an excellent paper from one of the leading groups in the field. They investigate brain dynamic in order to distinguish intrinsic and extrinsic dynamics using an arrow through time as borrowed from thermodynamics., They use intracranial recording data sets from non-human primates during three different states, awake, anesthesia, and sleep. They observe significant differences in the slow-fast hierarchy during the three different states which led them assume different balances of intrinsic vs extrinsic dynamics across the three states. This is an outstanding paper which raises only a few remarks.

- The abstract would be written in a more concrete and specific way...the first part is rather longish and may be shortened and replaced by more specific methodological details and results findings.

- I am not sure whether I understand the insideout concept the authors put forward in the introduction

- One key methodological issue is that they do not really probe any external stimulation; all is resting state albeit in different states.....hence their inference that the arrow of time reflects the system's external-internal equilibrium (as I understand it) it at best indirect.....

- Why not using task data for these conditions? That would provide direct testing of their key point that the arrow of time reflects environmental sensitivity of the system.....

- I am a little puzzled that their result part reads like a collation of figure caption...

- They mention timescales in abstract and introduction including their hierarchy but never come back to that in their results and findings..do I miss something here? Given that they investigate different conscious states, they may also want to consider the recent by Zilio et al. 2021 (Neuroimage) on timecales in different conscious states...

- The INSidEOUT is capitalized, why? The concept of inside-out is introduced by Buzsaki which is never mention here...the authors seem to take the insideout in a slightly different way but never really explicate that.....why do they use the same concept as Buszkai but capitlaized/ May be I missed something but I am not really sure whether they need this world-play....

- One thing I am missing is htat the authors investigate intrinsic vs extrinsic networks with their time arrow method...respectively...should they not behave reciprocally with respect of their time arrows?

- They speak of functional connectivity in figue 4...but I cannot find it in the method part? Did I miss something? Is it pahse-based coherence? Or how it calculated?

the same goes for the hierarchy notion in the same figure.....it seems to be a hierarchy of states within spaces but not a temporo-spatial hierarchy as they in the abstract...or am I wrong?

- The relation to consciousness may be at least briefly discussed...with mention of the leading theories like IIT, GNWT and TTC....

Reviewer #2 (Remarks to the Author):

In the manuscript titled "Inside out: Precise signatures of the balance of intrinsic and extrinsic dynamics in brain states" the authors present a data analysis technique for distinguishing between the balance of externally and internally driven brain states. They apply this method to electrophysiological non-human primate data recorded across different states of arousal, namely; awake, sleep, and several types of anaesthesia. The authors find their measure capable of differentiating these behaviourally distinct states, whereby brain dynamics during high arousal (wake and anaesthetic recovery) are bias to externally drivers and, conversely, anaesthetics are bias to internal drivers.

Overall the paper is well written, the results interesting and timely, and the conclusions appropriate. After sufficient response to my comments below, I recommend this manuscript for publication.

Key comments:

How was the time-shift (delay) selected? This forms a crucial part of the work but is not mentioned anywhere. Are the results shown here consistent over some range of delays? When does the difference in irreversibility become significant? How does this time horizon compare to the autocorrelation timescale?

How are the time resolved measures of irreversibility calculated for Figure 3(column 3) – since correlation is a time average?

How was the PCA calculated? Was the data across all sessions(and time direction) concatenated and the eigenmode decomposition performed on this data average covariance matrix? If the PCA was done uniquely to each data type the authors should show how the eigenvectors differ between each.

How do the main results depend on the number of included PCs?

Yanagawa et al re-referenced this ECOG data in their work. Did you also re-reference?

Minor:

L19: Fix grammar

L24: Fix grammar

L59: Fix grammar

L107: Please define the 'centre of the potential'

Figure 1: The notation in panel E and the methods section is inconsistent. ($y^{(r)}(t) \sim y(r)$, etc). Please make these consistent. Also, the interpretability of figures should be self-contained. As it stands a lot of the information for this figure lies in the methods. Thus, the overall utility of Figure 1 could be greatly improved by including more information in the figure caption.

L190: How do these results contrast the difference in the auto-correlation drop off across the hierarchy?

Figure 4: What underpins the low levels of spatial(hierarchical) variability of non-reversibility in the awake(and recovery) condition for Ketamine? This is significantly different to all other conditions.

Reviewer #3 (Remarks to the Author):

In this article, the authors propose to investigate the balance of intrinsic and extrinsic dynamics in brain states through quantifying the temporal asymmetry. This perspective is novel and interesting but there still remains several issues to address. I list my comments below.

1. The authors applied the reversibility to distinguish three brain states, awake, deep sleep, and anesthesia. From my experience, such difference would be easily to be identified through functional and effective connectivity. The additional insight beyond the connectivity should be emphasized explicitly for the current paper.
2. The balance between the intrinsic and extrinsic brain dynamics has been mentioned many times through the paper, can the authors provide a quantitative description about the following questions: 1) what's the mathematical/physical form of the balance? 2) how is the balance related to the time reversibility?
3. Based on the formula of $c_{forward}$, $c_{reversal}$, and $FS_{forward}$, $FS_{reversal}$, the proposed

measurements are very similar to the trend of time series. Thus I am a bit concerned that the significance is a bit overclaimed. Also, as the authors mentioned by themselves, the arrow of time is measured by the production of entropy. Thus the simplification to c_{forward} and etc. in the current paper requires justification both conceptually and mathematically.

4. The way of measuring the equilibrium is actually the stationarity of time series, which was discussed in many works before. The unique methodological value of the current work should be verified with more experiments.

5. There might be some inconsistency in the Figure 2A. The "eye open" and "wake" are not matched.

6. On page 7, the authors said that "This change in the hierarchy is driven by the breaking of the detailed balance associated with each condition." This should be verified by a causal analysis.

7. What are the statistical significance for the identified brain states and the hierarchical structures in addition to their differences? This might be verified by performing reproducible analysis on external dataset.

Reviewer #4 (Remarks to the Author):

In this paper the authors used a thermodynamic-inspired framework to quantify the balance between intrinsic and extrinsic dynamics in brain signals. They establish the temporal asymmetry of large-scale brain signals through an estimation of the arrow of time (i.e. difference between the shifted correlations of forward and reverse temporal order). They show that this allows to obtain precise, distinguishing signatures for three radically different brain states (awake, deep sleep and anaesthesia) in ECOG brain data from non-human primates. The approach is quantitatively rigorous, the methods are clear, and the results highlight some interesting features in the balance of intrinsic and extrinsic brain dynamics in sleep and anaesthesia brain states. I think this work makes a very interesting contribution. I only have few minor suggestions to improve this manuscript and increase its clarity.

- Methods. Time shift T : is it fixed, or is the framework being tested at different T ? Please clarify.

- Fig. 3. Are the Functional connectivity matrices PCA-denoised similarly to the INSIDEOUT components? They should be, in order to make them comparable with the INSIDEOUT framework..

- I really enjoyed reading the Discussion. I wonder whether the "inside out" balance of intrinsic and extrinsic dynamics could also serve as an individual "fingerprint", unique to each individual brain, since "brain fingerprints" seem to follow a similar sensory/higher-order regions hierarchy (see for instance "When makes you unique: Temporality of the human brain fingerprint", Van De Ville et al., 2021, Science Advances). What is the authors' take on this? Maybe relevant as a future direction?

- It's Erwin Schrödinger, not Ernst ;)

I have spotted a few typos (there might be more):

Line 20: This allows

Line 399-400 Too many "between"

Response to the reviewers of COMMSBIO-21-3486-T

We would like to thank you for your insightful and constructive comments, which have immensely strengthened our ms. We have systematically addressed all your concerns, commentaries and suggestions. As you can see below, we have performed many additional analyses including new relevant large-scale datasets using over 1000 human participants and resulted in six new supplementary figures. These substantial analyses and major revisions to the text have confirmed the results and helped clarify your insightful comments. We trust that these changes and clarifications mean that the revised ms is now suitable for publication.

Reviewers' comments:

Reviewer #1 (Remarks to the Author):

This is an excellent paper from one of the leading groups in the field. They investigate brain dynamic in order to distinguish intrinsic and extrinsic dynamics using an arrow through time as borrowed from thermodynamics., They use intracranial recording data sets from non-human primates during three different states, awake, anesthesia, and sleep. They observe significant differences in the slow-fast hierarchy during the three different states which led them assume different balances of intrinsic vs extrinsic dynamics across the three states. This is an outstanding paper which raises only a few remarks.

Thank you!

- The abstract would be written in a more concrete and specific way...the first part is rather longish and may be shortened and replaced by more specific methodological details and results findings.

Thank you for your constructive comment – we have modified the abstract, adding more information on methods and results.

- I am not sure whether I understand the insideout concept the authors put forward in the introduction

Thank you. In the revised ms, we have tried to better explain the key concept of the INSIDEOUT framework by adding more specific methodological information in the introduction.

- One key methodological issue is that they do not really probe any external stimulation; all is resting state albeit in different states.....hence their inference that the arrow of time reflects the system's external-internal equilibrium (as I understand it) it at best indirect.....

Thank you for raising this important issue, which we have now clarified in the revised ms. The point of INSIDEOUT framework is to quantify how there are different breaking of the detailed balance in different brain states. We define a change in the detailed balance of the brain dynamics as the change in the hierarchy of the causal interactions in the intrinsic brain dynamics, i.e. the breaking of the balance reflects the level of asymmetry of causal interactions. This asymmetry is captured through the arrow of time. As such we don't need to probe external stimulation since the unchanging environment is providing this stimulation and therefore a change in the hierarchy of causal interactions in brain dynamics must reflect the underlying different

brain states. Nevertheless, inspired by your comments (including the next one), we have also probed task stimulation...

- Why not using task data for these conditions? That would provide direct testing of their key point that the arrow of time reflects environmental sensitivity of the system.....

Thank you for this excellent idea. The public datasets provide ECOG data for different brain states in macaques. The available task data (movie and visual gratings) are passive (rather than active) tasks and do not contain baseline data. For this reason, we decided to test our INSIDEOUT framework on large-scale human data, contrasting resting baseline data with seven cognitive tasks. As you can imagine, this was a major undertaking and a major test for our framework, since BOLD data is different from ECOG data and humans are different from macaques. We are delighted to report that our INSIDEOUT framework works equally well in this unrelated dataset as shown now in a supplementary Figure S5 (and supplementary Methods).

- I am a little puzzled that their result part reads like a collation of figure caption...

Thank you, we have reworded the results section.

- They mention timescales in abstract and introduction including their hierarchy but never come back to that in their results and findings..do I miss something here? Given that they investigate different conscious states, they may also want to consider the recent by Zilio et al. 2021 (Neuroimage) on timescales in different conscious states...

We have now made sure that we do not mention timescale in the abstract and introduction. Nevertheless, thank you for highlighting this very interesting reference, which we have now incorporated into the discussion as part of our discussion of the relevance of timescale for the present results.

- The INSidEOUT is capitalized, why? The concept of inside-out is introduced by Buzsaki which is never mention here...the authors seem to take the insideout in a slightly different way but never really explicate that.....why do they use the same concept as Buszkai but capitlaized/ May be I missed something but I am not really sure whether they need this word-play....

Thank you for this very helpful comment. Truth be told, we were inspired in the naming by the eponymous Disney movie and were not aware of Buzsaki's excellent new book. We have now read, referenced and contextualised his ground-breaking ideas (which are much more wide-ranging than our aims here). In order to keep the two related ideas separate, rather than use Buzsaki's 'inside-out' naming, we would prefer to keep it capitalised and upper case 'INSIDEOUT', reflecting a specific instance of a broader concept.

- One thing I am missing is for the authors to investigate intrinsic vs extrinsic networks with their time arrow method...respectively...should they not behave reciprocally with respect to their time arrows?

Thank you for raising this important point. In the discussion, we have now clarified that we see the effect of the breaking of the detailed balance reflecting the asymmetry in the balance of intrinsic vs extrinsic dynamics in different brain states.

- They speak of functional connectivity in figure 4...but I cannot find it in the method part? Did I miss something? Is it phase-based coherence? Or how it calculated?

the same goes for the hierarchy notion in the same figure.....it seems to be a hierarchy of states within spaces but not a temporo-spatial hierarchy as they in the abstract...or am I wrong?

Thank you. In the Methods, we have added a description of how the functional connectivity measure shown in Figure 3 is simply the correlation of the timeseries of activity for a given brain state. Equally, we have expanded on the description of the hierarchy measure shown in Figure 4. This is computed as the standard deviation of the elements of the matrix FS_{diff} , which is the matrix of the squared difference between forward and reversed time-shifted correlations. In other words, this reflects the degree of asymmetries in the causal interactions between elements. As such this measures the spatiotemporal hierarchy.

- The relation to consciousness may be at least briefly discussed...with mention of the leading theories like IIT, GNWT and TTC....

Thank you for raising this very important point! We have updated the discussion to include an explicit link with these leading theories of consciousness.

Reviewer #2 (Remarks to the Author):

In the manuscript titled “Inside out: Precise signatures of the balance of intrinsic and extrinsic dynamics in brain states” the authors present a data analysis technique for distinguishing between the balance of externally and internally driven brain states. They apply this method to electrophysiological non-human primate data recorded across different states of arousal, namely; awake, sleep, and several types of anaesthesia. The authors find their measure capable of differentiating these behaviourally distinct states, whereby brain dynamics during high arousal (wake and anaesthetic recovery) are bias to externally drivers and, conversely, anaesthetics are bias to internal drivers.

Overall the paper is well written, the results interesting and timely, and the conclusions appropriate. After sufficient response to my comments below, I recommend this manuscript for publication.

Thank you!

Key comments:

How was the time-shift (delay) selected? This forms a crucial part of the work but is not mentioned anywhere. Are the results shown here consistent over some range of delays? When does the difference in irreversibility become significant? How does this time horizon compare to the autocorrelation timescale?

Thank you for pointing this out. We now specify T in the results and methods, and we specify that we selected the optimal T=4 in a two-step procedure, where we first compute the averaged autocorrelation over all signals and identified the approximate value of T, where the autocorrelation has sufficiently decayed. We then optimise around this value to find the most significant results.

How are the time resolved measures of irreversibility calculated for Figure 3(column 3) – since correlation is a time average?

We have clarified that the figures in column 3 are taken from representative single individuals in a given session.

How was the PCA calculated? Was the data across all sessions (and time direction) concatenated and the eigenmode decomposition performed on this data average covariance matrix? If the PCA was done uniquely to each data type the authors should show how the eigenvectors differ between each.

Thank you for asking for this important clarification, which prompted us to compute and compare different strategies of PCA. In the ms, we computed the PCA based on each session for each monkey in each condition. Note that the reversibility computation does not require a common reference. Nevertheless, to test your excellent point, we concatenated all the sessions for one monkey in awake and in sleep and compared the strategies of computing the reversibility on the PCA of independent sessions compared to concatenated sessions (common reference). We show and discuss the results in new supplementary Figure S2. As you can see, both strategies result in a similar level of significant differences in reversibility.

How do the main results depend on the number of included PCs?

Excellent question. We ran further analyses using different sizes of PCA to test the robustness of the results. As you can see in the new Figure S3, only using the small number of 4 PCAs gave non-significant results. Given that we use 10 components, we conclude that the main results do not depend on the number of PCAs (except for very small numbers).

Yanagawa et al re-referenced this ECOG data in their work. Did you also re-reference?

Thank you. We preferred to stay as close to the raw data as possible and therefore did not re-reference and instead used the same pre-processing as in Sanz Perl et al. 2021 and de la Fuente et al 2021. This is now clarified in the revised ms.

Minor:

L19: Fix grammar

Done.

L24: Fix grammar

Done.

L59: Fix grammar

Done.

L107: Please define the 'centre of the potential'

Done.

Figure 1: The notation in panel E and the methods section is inconsistent. ($\hat{y}(r)(t) \sim y(r)$, etc). Please make these consistent. Also, the interpretability of figures should be self-contained. As it stands a lot of the information for this figure lies in the methods. Thus, the overall utility of Figure 1 could be greatly improved by including more information in the figure caption.

We have made the notation in panel E consistent and have provided extra information to the figure caption of Figure 1.

L190: How do these results contrast the difference in the auto-correlation drop off across the hierarchy?

This is an interesting way of computing another hierarchical measure, however we are not sure that this would shed further light on the main question being investigated here, namely the hierarchy of reversibility rather than the hierarchy of the drop-off of the autocorrelation.

Figure 4: What underpins the low levels of spatial(hierarchal) variability of non-reversibility in the awake(and recovery) condition for Ketamine? This is significantly different to all other conditions.

Thank you for spotting this important error! Somehow the label of 2.5×10^{-3} became 2.5×10^{-4} . Fixed now.

Reviewer #3 (Remarks to the Author):

In this article, the authors propose to investigate the balance of intrinsic and extrinsic dynamics in brain states through quantifying the temporal asymmetry. This perspective is novel and interesting but there still remain several issues to address. I list my comments below.

Thank you!

1. The authors applied the reversibility to distinguish three brain states, awake, deep sleep, and anesthesia. From my experience, such difference would be easily to be identified through functional and effective connectivity. The additional insight beyond the connectivity should be emphasized explicitly for the current paper.

Excellent point. We now emphasise in the discussion that the INSIDEOUT framework is an alternative and complementary measure to effective connectivity and that it is much simpler than transfer entropy methods.

2. The balance between the intrinsic and extrinsic brain dynamics has been mentioned many times through the paper, can the authors provide a quantitative description about the following questions: 1) what's the mathematical/physical form of the balance? 2) how is the balance related to the time reversibility?

Thank you. Our framework is based on classical statistical physics which can establish the relationship between breaking the detailed balance and reversibility. We cite here this link as an interpretation of our results but the main aim here is to characterise the reversibility of brain states rather than establishing a mathematical relationship, since this link has already been established more formally in other research, see for example Sanz Perl et al, Phys Rev E (2021) and Lynn et al PNAS (2022). The former uses the flow of the system while the latter use spin models. We have added this to the discussion.

Please also see our answer to your excellent suggestion in point 6 below, where we used transfer entropy to compute a quantitative causal measure of the asymmetry and breaking of the detailed balance, which is consistent with the results of the INSIDEOUT

framework, and significantly strengthens the interpretation of our novel measure of reversibility.

3. Based on the formula of c_{forward} , c_{reversal} , and FS_{forward} , FS_{reversal} , the proposed measurements are very similar to the trend of time series. Thus I am a bit concern that the significance is a bit overclaimed. Also, as the authors mentioned by themselves, the arrow of time is measured by the production of entropy. Thus the simplification to c_{forward} and etc. in the current paper requires justification both conceptually and mathematically.

We thank you for raising this important point. You are absolutely right that this is a simplification of the true arrow of time in that instead of computing the production entropy, we use pairwise time-shifted correlations. However, this simplification is remarkably powerful as shown in the results. As you will know, it is very difficult to fully compute the production entropy of the brain. To the best of our knowledge there currently only two papers in the literature that estimates the production entropy (Lynn et al 2022, Sanz Perl et al 2021) but, in both cases, this is approximated through a low-dimensional space reduction. The advantage here is that our simpler framework can capture the arrow of time across all time signals, whether in ECOG or in fMRI. We have added this disclaimer to the discussion.

4. The way of measuring the equilibrium is actually the stationarity of time series, which was discussed in many works before. The unique methodological value of the current work should be verified with more experiments.

Thank you for this suggestion. In the revised ms, we have gone to considerable lengths to use the framework in large-scale HCP fMRI data of resting and seven tasks, which is now included in the supplementary material. As you can see, this broadens the unique methodological value and opens up for the use of this method to characterise human brain states directly in future work.

5. There might be some inconsistency in the Figure 2A. The “eye open” and “wake” are not matched.

Brilliant! Well-spotted, thank you. Corrected in figure and caption.

6. One page 7, the authors said that “This change in the hierarchy is driven by the breaking of the detailed balance associated with each condition.” This should be verified by a causal analysis.

Excellent suggestion! We have gone to considerable length to perform further causal analyses of the data using transfer entropy, which is now shown in the new Figure S4. As you can see this confirms the INSIDEOUT results given how the results are consistent with those provided using transfer entropy, which is a direct measure of causality.

7. What are the statistical significance for the identified brain states and the hierarchical structures in addition to their differences? This might be verified by performing reproducible analysis on external dataset.

Thank you. We have verified the INSIDEOUT framework not only on large-scale human HCP fMRI neuroimaging data but also on another fMRI dataset of human participants awake and in deep sleep. We have added a new Figure S6.

Reviewer #4 (Remarks to the Author):

In this paper the authors used a thermodynamic-inspired framework to quantify the balance between intrinsic and extrinsic dynamics in brain signals. They establish the temporal asymmetry of large-scale brain signals through an estimation of the arrow of time (i.e. difference between the shifted correlations of forward and reverse temporal order). They show that this allows to obtain precise, distinguishing signatures for three radically different brain states (awake, 29 deep sleep and anaesthesia) in ECG brain data from non-human primates. The approach is quantitatively rigorous, the methods are clear, and the results highlight some interesting features in the balance of intrinsic and extrinsic brain dynamics in sleep and anaesthesia brain states. I think this work makes a very interesting contribution.

Thank you!

I only have few minor suggestions to improve this manuscript and increase its clarity.

- Methods. Time shift T: is it fixed, or is the framework being tested at different T? Please clarify.

Thank you - we have clarified that we used a fixed T=4.

- Fig. 3. Are the Functional connectivity matrices PCA-denoised similarly to the INSIDEOUT components? They should be, in order to make them comparable with the INSIDEOUT framework.

Thank you for raising this point. We simply wanted to demonstrate that the functional connectivity matrices are less informative than the INSIDEOUT framework. We have now produced a new supplementary Figure S1, which shows the scatterplots of the variances in PCA space similar to those found in Figure 3, but with the same number of PCA=10 components as in the INSIDEOUT framework. As you can see, these scatterplots are not particularly useful for distinguishing brain states. In the revised ms, we added references to the new figure and made these points clearer.

- I really enjoyed reading the Discussion. I wonder whether the "inside out" balance of intrinsic and extrinsic dynamics could also serve as an individual "fingerprint", unique to each individual brain, since "brain fingerprints" seem to follow a similar sensory/higher-order regions hierarchy (see for instance "When makes you unique: Temporality of the human brain fingerprint", Van De Ville et al., 2021, Science Advances). What is the authors' take on this? Maybe relevant as a future direction?

Great idea! We added further discussions of how the INSIDEOUT framework could potentially serve as an individual 'fingerprint' unique to each individual brain, in particular in reference to this excellent paper, which we did not know.

- It's Erwin Schrödinger, not Ernst ;)

Of course! Corrected

I have spotted a few typos (there might be more):

Line 20: This allows

Line 399-400 Too many "between"

Corrected

REVIEWERS' COMMENTS:

Reviewer #1 (Remarks to the Author):

all my comments were addressed in an excellent way

Reviewer #2 (Remarks to the Author):

The authors have sufficiently addressed my concerns. I recommend this manuscript for publication.

Reviewer #3 (Remarks to the Author):

The authors have addressed my concerns and I recommend the manuscript to be accepted.